# Differentiating attack-defense performance for starting and bench players during the Tokyo Olympics men's basketball competition

Wenping Sun[1¤]*, ChenSoon Chee[1], LianYee Kok[2], FongPeng Lim[3], Shamsulariffin Samsudin[1]

1 Department of Sport Studies, Faculty of Educational Studies, Universiti Putra Malaysia, Kuala Lumpur, Malaysia, 2 Department of Sport Science, Faculty of Applied Sciences, Tunku Abdul Rahman University of Management and Technology, Kuala Lumpur, Malaysia, 3 Department of Mathematics and Statistics, Faculty of Science, Universiti Putra Malaysia, Kuala Lumpur, Malaysia

¤ Current address: Department of Sports, Nanyang Institute of Technology, Nanyang, China
* suncao850525@gmail.com

**Data Availability Statement:** All data for this study are available from the official website of the International Basketball Federation (FIBA) [https://www.fiba.basketball/olympics/men/2020]. Others

## Abstract

This study aimed to explore the differences in attack-defense performance between the top and bottom teams for starting and bench players during the Tokyo Olympics men's basketball competition, to determine the relationship between the attack-defense performance of starting and bench players and the final competition rankings, as well as with each performance indicator. The rank-sum ratio (RSR) comprehensive evaluation was employed to describe the attack-defense performance of starting and bench players. Additionally, an independent sample $t$-test, Spearman Rho Correlation, and Pearson Correlation were conducted to test the differences and relationships between the various variables at a 0.05 level of significance, respectively. The results indicated that the top four teams showed significant differences in the attack-defense performance of their starting players compared to the bottom four teams ($p = 0.021$), mainly in terms of 3-point shooting percentage ($p = 0.042$) and free throw shooting percentage ($p = 0.044$). Besides that, the attack-defense ranks of both starting players ($p = 0.004$, $r = 0.757$) and bench players ($p = 0.020$, $r = 0.658$) had a significant correlation with the final rankings. Points per game, 2-point field goal percentage, and assists had a statistically significant ($p < 0.01$) and strong positive correlation ($0.70 < r < 0.90$) with the attack-defense performance of both starting and bench players. 3-point field goal percentage, offensive rebounds, defensive rebounds, steals, and blocks were the technical indicators that distinguish starting from bench players. In conclusion, one of the common characteristics of the top national basketball teams was the strong attack-defense ability of the starting players. It is recommended that coaches select players with stronger 3-point shooting ability and more accurate free-throw shooting into the team's starting rotation.

## Introduction

Basketball is a complex, high-scoring team sport with high-intensity intermittent characteristics [1, 2]. The game events and situations that arise during the game are dynamic and require

would be able to access or request these data in the same manner as the authors, with no special access or request privileges.

**Funding:** The author(s) received no specific funding for this work.

**Competing interests:** The authors have declared that no competing interests exist.

timely responses from coaches based on the on-court competitive demands [1]. Especially when there are more fouls, poor performance, fatigue, or injuries to players on the court, the coach needs to rotate players and adjust tactics in a timely manner to maintain the team's fighting ability [1, 3]. Therefore, proper player rotation is one of the most essential strategies for coaches to prepare for games and respond to changes in on-court situations [3], in order to optimize the overall performance of the team to win games [1].

As we all know, the role of players in basketball is divided into starting players and bench players [2, 4]. Unlike a limited number of rotations in soccer or rugby, unlimited substitutions can be made in basketball [1, 5]. In recent years, the increased intensity of high-level competitive basketball has also led to an increase in the number of player rotations, especially in the NBA and European Basketball League [6]. As a result, the contribution of bench players, or "a deeper bench", was considered one of the key factors in winning games in elite basketball competitions [7–10]. Based on the importance of bench players in the basketball game, many scholars have conducted studies on starting players and bench players from different perspectives, such as game demands [2, 4], training load [11, 12], aerobic capacity [13], etc.

As a team sport, the basketball game contains three phases: attack, defense, and transition [14–17]. Attack and defense are two basic forms of basketball games [18–21], and the attacking and defensive ability of a team or player can be reflected by quantifying game-related performance indicators such as rebounds, assists, and steals [22–27]. However, to the best of our knowledge, there were only a very limited number of studies that differentiated starting and bench players through different game-related indicators. Among them, one study proposed that fouls and defensive rebounds were the most important performance indicators to distinguish starting and bench players in the Portuguese men's professional basketball league [28]. Another study with different results stated that successful 2-point shooting, free throw percentage, and assists were the indicators that differentiated between the two types of players in the Women's National Basketball Association league (WNBA) [10]. However, there was no related research integrating these performance indicators to reflect the overall attack-defense ability of starting and bench players, as well as the relationship between their attack-defense performance and the game outcome. Most relevant studies examined differences between the two groups of starting and bench players in basketball teams [2, 4, 10–13, 28]. Currently, there is a lack of scientific literature focused on examining the attacking and defensive performance between starting players of different national basketball teams (especially the top and bottom teams) during the elite level competition, as well as between bench players.

Therefore, this study was designed to analyze and evaluate the attack-defense performance of each team's starting and bench players during the Tokyo Olympics men's basketball competition. Also, the analysis specifically focused on starting and bench players from the top four and bottom four teams in this competition. This study aimed to explore the differences in attack-defense performance between the top and bottom teams for starting and bench players, as well as discover common characteristics of the top basketball teams. Also, to determine the relationship between the attack-defense performance of the starting and bench players and the final competition rankings, as well as with each attack-defense performance indicator. It was hypothesized that there were significant differences in attack-defense performance between the top four and bottom four teams for starting players, as well as for bench players, and significant differences in each attack-defense performance indicator. In addition, the attack-defense performance of both starting and bench players was significantly related to the team's final ranking as well as to each attack-defense performance indicator.

## Materials and methods

### Sample

The Tokyo Olympics Men's Basketball Tournament consisted of 12 teams: the United States of America (USA), France, Australia, Slovenia, Italy, Spain, Argentina, Germany, the Czech Republic, Nigeria, Japan, and Iran. In this study, the sample comprised 144 male players from 12 participating teams, which were divided into two groups (i.e., starting and bench players) according to their roles on the field. The starting players for each game were the five players who started the game on the court, and the rest of the players were bench players [2]. The role of each team's player (starting or bench player) was obtained from the FIBA webpage [29], where each player was marked as a starting or bench player in the box-score after each game. This study was conducted based on the game-related statistics of starting and bench players, involving a total of 26 matches from the first round to the final.

### Data collection

All data for this study was collected from the basketball database on the International Basketball Federation's official website [29]. In basketball, the box-score is a frequently evaluated database used to collect data about player performance. Generally, the attacking performance indicators include points per game (PPG), 2-point shoot percentage (2P%), 3-point shoot percentage (3P%), free throws percentage (FT%), offensive rebounds (OR), assists (As), and turnovers (To) in the basketball game [30], while the defensive performance indicators include defensive rebounds (DR), steals, blocks, and fouls [31, 32]. In this study, the attacking and defensive performance indicators were combined to reflect the overall attack-defense performance of starting and bench players [30, 32].

The process of data collection for each team was performed in the following ways: (i) identification of starting and bench players in the game according to the information provided on the box-score page for each game on the FIBA website, (ii) extraction of game-related scores of each player for each performance indicator, (iii) calculating each player's mean score for the competition (total score for each indicator divided by the number of matches played), (iv) calculating the mean score for each performance indicator according to starting and bench players. In addition, other variables such as rebounds, assists, steals, blocks, fouls, and turnovers were analyzed from video recordings of each match, and the recorded data were compared with FIBA data for verification of data accuracy and consistency.

### Data analysis

After the mean scores for all attacking and defending indicators according to starting and bench players have been determined, the rank-sum ratio (RSR) comprehensive evaluation was used to calculate the ranking of each team for every performance indicator [18, 20, 33]. The RSR calculation formula is denoted as RSR = $\sum R/(M*N)$, where R refers to the rank value of each evaluation index, $\sum R$ refers to the rank sum of all evaluation indexes for a team, M refers to the number of evaluation indexes, and N refers to the number of teams. As for indexes that are better when they are higher, R values should be coded from small to large, while indexes that are better when they are lower should be coded from large to small. When the ranks of some indexes are the same, the average of these indexes normal rank is taken [34]. RSR values are between 0 and 1, with higher values indicative of better performance [30, 33]. The 5-level evaluation criteria of the RSR were used in this study (Table 1) with RSR values greater than or equal to 0.8 denoting a very strong attack-defense performance (category A) with decreasing RSR values indicating strong (category B, 0.60–0.79), moderate (category C, 0.40–0.59), weak

**Table 1. The RSR comprehensive evaluation criteria.**

| A | B | C | D | E |
|---|---|---|---|---|
| ≥0.8 | 0.60~0.79 | 0.40~0.59 | 0.20~0.39 | ≤0.19 |
| Very Strong | Strong | Moderate | Weak | Very Weak |

Note: Percentile of Rank-Sum Ratio (RSR) suggested by Tian Fengdiao.

(category D, 0.20–0.39), and very weak (category E, ≤0.19) attack-defense performance, respectively.

According to the calculation formula of the RSR comprehensive evaluation, the attack-defense RSR value of starting and bench players for each team can be obtained by quantifying and synthesizing the specific attack-defense performance indicators. Subsequently, the attack-defense performance of starting and bench players for each team can be described based on the RSR comprehensive evaluation criteria, as well as the disparity between them. Furthermore, the advantages and disadvantages in the attack-defense performance of starting and bench players for each team can also be described according to the rank and value of specific performance indicators.

In addition, to obtain information regarding common and specific characteristics of starting and bench players among the top teams, as well as to examine any differences in attack-defense performance indicators for starting and bench players between the top and bottom teams, the 12 participating teams were divided into three groups according to their final ranking: the top four, the middle four, and the bottom four. This format of separating the top and bottom performing teams with a "middle performing" group is quite commonly used to differentiate performance characteristics between different levels of ability as there is usually no obvious difference if two groups with similar rankings are selected for comparison.

The IBM SPSS 25.0 software (The IBM Corporation) was used for the inferential analysis. An independent sample *t*-test was conducted to test the difference in attack-defense performance of stating and bench players between the top four and bottom four teams, as well as each performance indicator including PPG, 2P %, 3P %, FT %, offensive rebounds, assists, turnovers, defensive rebounds, steals, blocks, and fouls. The Shapiro-Wilk test showed that the dependent variables used for the independent sample *t*-test were normally distributed. Also, Levene's test verified the homogeneity of the variances of the dependent variables. Additionally, Spearman Rho Correlation and Pearson Correlation Analysis were performed at the 0.05 significance level to obtain the correlation coefficients between various variables, and then the strength of the relationship between the variables was determined based on the Guildford Rule of Thumb. The Spearman Rho Correlation was used to determine the relationship between the attack-defense performance rankings of starting and bench players and the final competition rankings, respectively. Pearson Correlation was employed to test the relationship between the attack-defense RSR value of starting and bench players and performance indicators. Based on these correlation analyses, it was possible to determine the contribution of various performance indicators to the attack-defense performance of starting and bench players, as well as the impact of the attack-defense performance of starting and bench players on the final competition rankings.

## Results

### The RSR comprehensive evaluation on attack-defense performance for starting players among 12 participating teams

According to the calculation formula of the RSR comprehensive evaluation, the attack-defense RSR values of starting players for each team were obtained (Table 2). As can be seen from

Table 2, the starting players of the champion USA had the strongest attack-defense ability among the 12 participating teams, with an RSR value of 0.72, which belonged to class B level. Followed by Slovenia, Australia, and France, with RSR values of 0.69, 0.67, and 0.63, respectively, all of which also belonged to class B level. Italy, the Czech Republic, Japan, Spain, Iran, Argentina, and Germany all belonged to class C level, with RSR values of 0.58, 0.56, 0.56, 0.50, 0.46, 0.44, and 0.40, respectively. Nigeria's starting players were the weakest in attack-defense ability, with an RSR value of 0.28, the only team that belonged to class D level. However, it was worth noting that no team's starting players reached class A level in the attack-defense ability, indicating that even the top team's starters may have shortcomings in some performance indicators, such as the USA team's free throw shooting percentage, offensive rebounds, and defensive rebounds.

## The RSR comprehensive evaluation on attack-defense performance for bench players among 12 participating teams

The attack-defense performance indicators of bench players among 12 participating teams were assigned according to the principle of RSR comprehensive evaluation, and finally, the RSR value and grade of bench players for each team were calculated (Table 3).

Like the starting players, the USA team's bench players had the best attack-defense performance among all participating teams, with an RSR of 0.72, which belonged to class B level. Followed by Australia, Nigeria, and Spain, with RSR values of 0.68, 0.66, and 0.64, respectively, all of which also belonged to class B level. Italy, France, Germany, Slovenia, Argentina, the Czech Republic, and Japan all belonged to class C level, with RSR values of 0.58, 0.52, 0.52, 0.51, 0.48, 0.47, and 0.40, respectively. The bench players of the Iranian team, which ranked last, had the worst attack-defense performance, with an RSR value of 0.33, which only belonged to class D level. Similarly, no team's bench players can reach the class A level in attack-defense ability, as each team's bench players had some shortcomings in their attack-defense performance indicators.

## The differences in attack-defense performance for starting and bench players between the top four and bottom four teams

The top four and bottom four teams were used as independent variables, while the attack-defense performance (i.e., RSR values) of the starting and bench players were used as

**Table 2. The RSR value and grade of the attack-defense performance for starting players among 12 participating teams.**

| FR | Teams | PPG | R | 2P % | R | 3P % | R | FT % | R | OR | R | As | R | To | R | DR | R | Steals | R | Blocks | R | Fouls | R | RSR | Grade | Rank |
|----|-------|-----|---|------|---|------|---|------|---|----|---|----|---|----|---|----|---|--------|---|--------|---|-------|---|-----|-------|------|
| 1 | USA | 57.3 | 8 | 59.2 | 11 | 37.8 | 9 | 77.2 | 5 | 5.9 | 6 | 14.3 | 9 | 6.0 | 12 | 15.7 | 6 | 6.0 | 10 | 3.7 | 12 | 10.5 | 7 | 0.72 | B | 1 |
| 2 | France | 58.3 | 9 | 53.2 | 7 | 38.8 | 12 | 72.7 | 7 | 6.2 | 8 | 14.0 | 7.5 | 8.8 | 6 | 20.5 | 11 | 3.8 | 1.5 | 2.0 | 8.5 | 11.0 | 6 | 0.63 | B | 4 |
| 3 | Australia | 60.8 | 10 | 51.3 | 5 | 38.7 | 11 | 72.4 | 6 | 6.8 | 11 | 17.5 | 12 | 8.0 | 8.5 | 13.8 | 3 | 5.7 | 8 | 1.5 | 5 | 9.3 | 8.5 | 0.67 | B | 3 |
| 4 | Slovenia | 70.3 | 12 | 63.6 | 12 | 34.7 | 6 | 72.0 | 4 | 8.2 | 12 | 16.0 | 10.5 | 8.7 | 7 | 23.3 | 12 | 4.3 | 4 | 2.0 | 8.5 | 11.7 | 2.5 | 0.69 | B | 2 |
| 5 | Italy | 51.5 | 5 | 55.4 | 9 | 35.6 | 8 | 78.8 | 10 | 5.5 | 5 | 10.0 | 3 | 6.5 | 11 | 11.8 | 2 | 6.0 | 10 | 1.3 | 4 | 8.8 | 10 | 0.58 | C | 5 |
| 6 | Spain | 48.8 | 4 | 54.9 | 8 | 37.9 | 10 | 83.3 | 12 | 3.3 | 2 | 11.5 | 4 | 7.0 | 10 | 15.8 | 7 | 4.0 | 3 | 1.0 | 2.5 | 11.5 | 4 | 0.50 | C | 8 |
| 7 | Argentina | 56.0 | 7 | 52.6 | 6 | 31.5 | 3 | 79.2 | 11 | 5.0 | 3.5 | 12.5 | 5 | 9.0 | 4.5 | 16.3 | 9 | 5.5 | 7 | 0.8 | 1 | 11.8 | 1 | 0.44 | C | 10 |
| 8 | Germany | 43.5 | 2 | 46.0 | 2 | 34.8 | 7 | 76.7 | 9 | 6.5 | 9 | 9.8 | 2 | 9.5 | 3 | 14.3 | 4 | 3.8 | 1.5 | 1.0 | 2.5 | 8.0 | 11 | 0.40 | C | 11 |
| 9 | Czech | 47.3 | 3 | 57.5 | 10 | 30.0 | 2 | 55.6 | 1 | 6.7 | 10 | 16.0 | 10.5 | 10.3 | 2 | 14.7 | 5 | 7.7 | 12 | 1.7 | 6 | 6.3 | 12 | 0.56 | C | 6 |
| 10 | Nigeria | 29.7 | 1 | 45.1 | 1 | 21.6 | 1 | 57.6 | 2 | 2.3 | 1 | 7.3 | 1 | 8.0 | 8.5 | 11.0 | 1 | 6.0 | 10 | 2.0 | 8.5 | 11.7 | 2.5 | 0.28 | D | 12 |
| 11 | Japan | 63.3 | 11 | 48.7 | 3 | 33.3 | 4 | 74.1 | 8 | 6.0 | 7 | 13.3 | 6 | 9.0 | 4.5 | 17.0 | 10 | 4.7 | 5 | 2.7 | 11 | 11.3 | 5 | 0.56 | C | 6 |
| 12 | Iran | 53.0 | 6 | 50.0 | 4 | 34.1 | 5 | 63.0 | 3 | 5.0 | 3.5 | 14.0 | 7.5 | 13.3 | 1 | 16.0 | 8 | 5.3 | 6 | 2.0 | 8.5 | 9.3 | 8.5 | 0.46 | C | 9 |

Note: FR (The final ranking during Tokyo Olympics men's basketball game), PPG (Points per Game), 2P% (2-point field goal made percentage), 3P% (3-point field goal made percentage), FT% (Free throws field goal made percentage), OR (Offensive Rebounds), As (Assists), To (Turnovers), DR (Defensive Rebounds).

**Table 3. The RSR value and grade of the attack-defense performance for bench players among 12 participating teams.**

| FR | Teams | PPG | R | 2P % | R | 3P % | R | FT % | R | OR | R | As | R | To | R | DR | R | Steals | R | Blocks | R | Fouls | R | RSR | Grade | Rank |
|----|-------|-----|---|------|---|------|---|------|---|-----|---|-----|---|-----|---|------|-----|--------|-----|--------|-----|-------|------|------|-------|------|
| 1 | USA | 41.7 | 11 | 60.0 | 12 | 43.2 | 10 | 79.1 | 6 | 3.0 | 6 | 10.0 | 12 | 4.2 | 9 | 10.0 | 8.5 | 3.8 | 10 | 1.5 | 9 | 9.8 | 2 | 0.72 | B | 1 |
| 2 | France | 27.5 | 4 | 57.7 | 10 | 34.6 | 5 | 80.8 | 9 | 2.3 | 3.5 | 8.0 | 10 | 5.5 | 4 | 10.0 | 8.5 | 2.3 | 5.5 | 0.5 | 5.5 | 9.3 | 3.5 | 0.52 | C | 6 |
| 3 | Australia | 29.0 | 5 | 56.7 | 9 | 42.3 | 9 | 80.0 | 7.5 | 3.2 | 7 | 7.8 | 9 | 4.3 | 7.5 | 9.3 | 7 | 4.2 | 11 | 0.7 | 7 | 6.7 | 10.5 | 0.68 | B | 2 |
| 4 | Slovenia | 30.5 | 6 | 55.7 | 8 | 39.7 | 8 | 70.0 | 5 | 3.5 | 9 | 6.7 | 5 | 3.7 | 10 | 8.0 | 4.5 | 1.8 | 3 | 0.0 | 1.5 | 8.8 | 7.5 | 0.51 | C | 8 |
| 5 | Italy | 31.0 | 7 | 46.7 | 3 | 25.5 | 2 | 88.9 | 10.5 | 2.3 | 3.5 | 6.8 | 6.5 | 2.3 | 11 | 11.8 | 10 | 2.5 | 7.5 | 1.8 | 10 | 9.0 | 5.5 | 0.58 | C | 5 |
| 6 | Spain | 35.5 | 9 | 58.5 | 11 | 29.1 | 3 | 69.2 | 4 | 5.0 | 12 | 6.8 | 6.5 | 6.5 | 2.5 | 16.7 | 12 | 2.3 | 5.5 | 2.5 | 12 | 8.8 | 7.5 | 0.64 | B | 4 |
| 7 | Argentina | 25.8 | 3 | 50.7 | 5 | 23.7 | 1 | 88.9 | 10.5 | 4.3 | 10.5 | 5.8 | 3 | 5.3 | 5.5 | 8.0 | 4.5 | 2.8 | 9 | 0.5 | 5.5 | 9.0 | 5.5 | 0.48 | C | 9 |
| 8 | Germany | 38.3 | 10 | 54.1 | 6 | 47.5 | 11 | 89.7 | 12 | 1.8 | 2 | 6.0 | 4 | 6.5 | 2.5 | 8.8 | 6 | 2.5 | 7.5 | 0.3 | 3.5 | 9.3 | 3.5 | 0.52 | C | 6 |
| 9 | Czech | 34.3 | 8 | 54.2 | 7 | 37.1 | 6 | 60.0 | 2 | 2.7 | 5 | 7.3 | 8 | 4.3 | 7.5 | 6.3 | 2 | 2.0 | 4 | 0.3 | 3.5 | 7.7 | 9 | 0.47 | C | 10 |
| 10 | Nigeria | 47.0 | 12 | 49.0 | 4 | 52.9 | 12 | 55.6 | 1 | 4.3 | 10.5 | 9.0 | 11 | 12.3 | 1 | 15.3 | 11 | 5.0 | 12 | 2.3 | 11 | 11.7 | 1 | 0.66 | B | 3 |
| 11 | Japan | 15.0 | 1 | 31.8 | 1 | 37.5 | 7 | 80.0 | 7.5 | 1.7 | 1 | 3.7 | 1 | 1.7 | 12 | 5.0 | 1 | 1.0 | 1 | 1.3 | 8 | 5.3 | 12 | 0.40 | C | 11 |
| 12 | Iran | 15.7 | 2 | 33.3 | 2 | 31.8 | 4 | 66.7 | 3 | 3.3 | 8 | 4.0 | 2 | 5.3 | 5.5 | 7.7 | 3 | 1.3 | 2 | 0.0 | 1.5 | 6.7 | 10.5 | 0.33 | D | 12 |

Note: FR (The final ranking during Tokyo Olympics men's basketball game), PPG (Points per Game), 2P% (2-point field goal made percentage), 3P% (3-point field goal made percentage), FT% (Free throws field goal made percentage), OR (Offensive Rebounds), As (Assists), To (Turnovers), DR (Defensive Rebounds).

dependent variables, respectively. The results of the independent sample $t$-test showed the top four teams had significant differences in the attack-defense performance of the starting players compared to the bottom four teams ($p = 0.021$), while the bench players did not show significant differences ($p = 0.161$).

## The differences in attack-defense performance indicators between the top four and bottom four teams for the starting and bench players

The top four and bottom four teams were used as independent variables, and each attack-defense performance indicator including PPG, 2P %, 3P %, FT %, offensive rebounds, assists, turnovers, defensive rebounds, steals, blocks, and fouls for the starting and bench players were used as dependent variables, respectively (Table 4).

The results of the independent sample $t$-test showed that the starting players of the top four teams performed better than the bottom four teams in terms of PPG, 2P%, 3P%, FT%, offensive rebounds, assists, turnovers, defensive rebounds, and blocks. There was a significant difference in 3P% ($p = 0.042$) and FT% ($p = 0.044$) between the top four and bottom four teams for the starting players, but no significant difference in other performance indicators ($p>0.05$). It was worth noting that the starting players of the bottom four teams outperformed the top four teams in terms of steals and fouls, but there was no significant difference ($p>0.05$). In addition, a comparison between the bench players of the top four and bottom four teams revealed that only 2P% showed significant differences ($p = 0.035$), while other performance indicators did not show significant differences ($p>0.05$), and even the bottom four teams outperformed the top four teams in terms of blocks and fouls.

**Table 4. The differences in attack-defense performance indicators between the top four and bottom four teams for the starting and bench players.**

| | PPG | 2P % | 3P % | FT % | OR | As | To | DR | Steals | Blocks | Fouls |
|---|---|---|---|---|---|---|---|---|---|---|---|
| **The starting players** | | | | | | | | | | | |
| Top four (x ±S$_d$) | 61.7 ±5.9 | 56.8 ±5.6 | 37.5 ±1.9 | 73.8 ±2.4 | 6.8 ±1.0 | 15.5 ±1.6 | 7.9 ±1.3 | 18.3 ±4.4 | 5.0 ±1.1 | 2.3 ±1.0 | 10.6 ±1.0 |
| Bottom four (x±S$_d$) | 48.3 ±14.1 | 50.3 ±5.2 | 29.8 ±5.7 | 62.6 ±8.3 | 5.0 ±1.9 | 12.7 ±3.7 | 10.2 ±2.3 | 14.7 ±2.6 | 5.9 ±1.3 | 2.1 ±0.4 | 9.7 ±2.5 |
| Difference | 13.4 | 6.5 | 7.7 | 11.2 | 1.8 | 2.8 | -2.3 | 3.6 | -0.9 | 0.2 | 0.9 |
| T | 1.748 | 1.694 | 2.571 | 2.545 | 1.626 | 1.371 | -1.721 | 1.436 | -1.161 | 0.380 | 0.731 |
| $P$ (2-tailed) | 0.131 | 0.141 | 0.042* | 0.044* | 0.155 | 0.219 | 0.136 | 0.201 | 0.290 | 0.717 | 0.492 |
| **The bench players** | | | | | | | | | | | |
| | PPG | 2P % | 3P % | FT % | OR | As | To | DR | Steals | Blocks | Fouls |
| Top four (x ±S$_d$) | 32.2 ±6.5 | 57.5 ±1.8 | 40.0 ±3.9 | 77.5 ±5.0 | 3.0 ±0.5 | 8.1 ±1.4 | 4.4 ±0.8 | 9.3 ±0.9 | 3.0 ±1.2 | 0.7 ±0.6 | 8.7 ±1.4 |
| Bottom four (x±S$_d$) | 28.0 ±15.5 | 42.1 ±11.2 | 39.8 ±9.1 | 65.6 ±10.6 | 3.0 ±1.1 | 6.0 ±2.6 | 5.9 ±4.5 | 8.6 ±4.6 | 2.3 ±1.8 | 1.0 ±1.0 | 7.9 ±2.7 |
| Difference | 4.2 | 15.4 | 0.2 | 11.9 | 0.0 | 2.1 | -1.5 | 0.7 | 0.7 | -0.3 | 0.8 |
| T | 0.497 | 2.718 | 0.025 | 2.021 | 0.000 | 1.454 | -0.642 | 0.318 | 0.646 | -0.493 | 0.521 |
| $P$ (2-tailed) | 0.637 | 0.035* | 0.981 | 0.090 | 1.000 | 0.196 | 0.544 | 0.761 | 0.542 | 0.639 | 0.621 |

Note

* $p<0.05$, with a significant difference. PPG (Points per Game), 2P% (2-point field goal made percentage), 3P% (3-point field goal made percentage), FT% (Free throws field goal made percentage), OR (Offensive Rebounds), As (Assists), To (Turnovers), DR (Defensive Rebounds).

**Table 5. Correlation between the attack-defense RSR ranks of starting and bench players and the final rankings.**

|  | Starting players | Bench players |
|---|---|---|
| Spearman's rho | 0.757** | 0.658* |
| Sig.(2-tailed) | 0.004 | 0.020 |
| N | 12 | 12 |

Note

* Correlation is significant at the 0.05 level (2-tailed)

** Correlation is significant at the 0.01 level (2-tailed).

## The correlation between the attack-defense RSR ranks of starting and bench players and the final rankings

The attack-defense RSR ranks of the starting and bench players among 12 participating teams were used as independent variables, respectively, and the final rankings for each team during the Tokyo Olympics men's basketball game were used as dependent variables. The correlation coefficients between the independent variables and dependent variables were obtained after running the Spearman Rho correlation analysis in Table 5.

It can be seen from Table 5, both starting players ($p = 0.004$) and bench players ($p = 0.020$) had a significant relationship with the final rankings. However, based on Guildford Rule of Thumb, the attack-defense RSR ranks of starting players had a positive and high correlation with the final rankings ($r = 0.757$), while bench players had a positive and moderate correlation ($r = 0.658$).

## The correlation between the attack-defense performance of starting and bench players and performance indicators

The attack-defense performance indicators were used as the independent variables, and the attack-defense RSR values of starting and bench players among the 12 participating teams were used as the dependent variables, respectively. The correlation coefficients between the independent variables and dependent variables were obtained after running Pearson correlation analysis in Table 6.

As shown in Table 6, PPG,2P%,3P%, offensive rebounds, and assists had a significant relationship with the attack-defense performance of the starting players ($p<0.01$), showing a positive and high relationship with correlation coefficients of 0.808, 0.751, 0.723, 0.707, and 0.769, respectively. In addition, for the bench players, PPG, 2P%, assists, defensive rebounds, steals, and blocks had a significant relationship with the attack-defense performance ($p<0.05$). Among them, PPG, 2P%, assists, and steals showed a positive and high relationship with correlation coefficients of 0.775, 0.715, 0.854, and 0.823, respectively, whereas defensive rebounds and blocks showed a positive and moderate relationship with correlation coefficients of 0.662 and 0.606, respectively. It was worth noting that PPG, 2P%, and assists had a significant relationship with the attack-defense performance of both starting and bench players ($p<0.01$), all showing a positive and high relationship ($0.70<r<0.90$).

## Discussion

Based on the present study, the USA team had the best attack-defense performance among all participating teams, whether they were starting players or bench players, and probably this was an important reason why they won the final championship. In addition, there was a significant difference in attack-defense performance between the top four and bottom four teams for the

**Table 6. Correlation between the attack-defense performance of starting and bench players and performance indicators.**

| Indicators | Attack-defense RSR of starting players | | Attack-defense RSR of bench players | |
|---|---|---|---|---|
| | Pearson (*r*) | Sig.(2-tailed) | Pearson (*r*) | Sig.(2-tailed) |
| PPG | 0.808** | 0.001 | 0.775** | 0.003 |
| 2P % | 0.751** | 0.005 | 0.715** | 0.009 |
| 3P % | 0.723** | 0.008 | 0.361 | 0.249 |
| FT % | 0.293 | 0.355 | 0.004 | 0.990 |
| OR | 0.707** | 0.010 | 0.327 | 0.300 |
| As | 0.769** | 0.003 | 0.854** | 0.000 |
| To | -0.318 | 0.313 | 0.317 | 0.316 |
| DR | 0.514 | 0.087 | 0.662* | 0.019 |
| Steals | 0.011 | 0.974 | 0.823** | 0.001 |
| Blocks | 0.450 | 0.142 | 0.606* | 0.037 |
| Fouls | -0.039 | 0.904 | 0.563 | 0.057 |

Note

\* Correlation is significant at the 0.05 level (2-tailed)

\*\* Correlation is significant at the 0.01 level (2-tailed). PPG (Points per Game), 2P% (2-point field goal made percentage), 3P% (3-point field goal made percentage), FT% (Free throws field goal made percentage), OR (Offensive Rebounds), As (Assists), To (Turnovers), DR (Defensive Rebounds).

starting players but not for the bench players. In terms of attack-defense performance indicators, there was a significant difference in terms of 3P% and FT% for the starting players between the top four and bottom four teams, and 2P% for the bench players, while there was no significant difference in the rest of the performance indicators. Therefore, it can be concluded that players' shooting percentages were the main technical indicators that distinguish high-level teams from low-level teams. Besides that, the attack-defense RSR ranks of starting players had a high correlation with the final rankings, while bench players had a moderate correlation. PPG, 2P%, and assists had a very significant and positive correlation with the attack-defense performance of both starting and bench players. 3P%, offensive rebounds, defensive rebounds, steals, and blocks were the technical indicators that distinguish starting from bench players.

## The correlation between the attack-defense performance of starting and bench players and the final competition rankings

Top basketball teams, whether they are national teams or club teams, tend to be strong on both the attack and defense ends, as has been reported in previous studies [35]. As this study showed, the attack-defense performance of both starting and bench players has a significant and positive relationship with the team's final game ranking. Similar results were found in studies of the 2014 Men's Basketball World Cup [32], Brazil Olympic men's basketball tournament [30], and the London Olympic women's basketball tournament [36]. However, one study stated that only defensive performance had a significant relationship with the final competition rankings during the 2015 Asian Men's Basketball Championship [33]. In international high-level men's basketball tournaments such as the Olympics or the Men's World Cup, we can see that often European and American teams are stronger, while Asian and African teams are weaker, often lose by large margins, and end up at the bottom of the rankings [9]. Therefore, the reason for the different research results may be caused by the lower level of Asian

men's basketball, which is not in line with the trend of world basketball development. For the bottom-ranked national teams, they should pay more attention to the balanced development of attack and defense in the future and further improve the overall attack-defense ability of both starting and bench players.

## Performance indicators that discriminate between starting and bench players

There were very few studies that used game-related statistics to discriminate starting and bench players [10, 32, 35]. Among them, some studies proposed that fouls and defensive rebounds were the most important performance indicators to distinguish starting and bench players in the Portuguese men's professional basketball league [28]. However, the present study agreed that defensive rebounds were one of the performance indicators that distinguish starting and bench players in the Tokyo Olympics men's basketball tournament, but fouls were not. As evidenced in some studies of national basketball leagues, bench players in club teams have lower aerobic capacity compared to the starting players so that they fatigue more easily during the game which leads to more fouls [5, 13], and also, as a result of a lack of confidence in their own performances, the bench players may not be as prepared to play defense as the starters, which leads to a higher number of fouls [28]. However, the various national teams participating in the Olympics are undoubtedly made up of the top players in their respective countries [9], and it can be argued that they should be similar in terms of aerobic capacity or self-confidence, unlike the starting and bench players in clubs where there was some disparity, a hypothesis that is expected to be confirmed in future studies. Therefore, this may be the reason that fouls were not the performance indicator for differentiating between the starting and bench players on the national teams in this study. Moreover, not only defensive rebounding, but offensive rebounding was also a performance indicator distinguishing starting players from bench players in this study. The reason for this difference may be related to the bouncing ability of the players, as it has been found that starting players tend to jump better and have a faster release [37]. Alternatively, it could be due to differences in players' technical skills in pivoting, blocking, anticipation, securing, and pulling the ball away [28].

The 3-pointers have a longer shooting distance than the 2-pointers, which required players to have more accurate shooting ability, so the 3-point shooting percentage was also a performance indicator to distinguish starting and bench players in this study. This result was confirmed by a study of NCAA men's basketball in the United States which indicated that bench players were more clearly biased toward shooting three-pointers in the game, but the success rate often showed significant differences from the starting players [38]. In terms of kinematic characteristics, the three-point shot requires greater release height and vertical displacement, as well as lower release angles, compared to the two-point shot, which may improve the shooting accuracy [39, 40]. Therefore, bench players may also need to pay attention to this in future training. In addition, two defensive indicators, steals and blocks, can also distinguish between starters and bench players, which can be attributed to the fact that starting players have greater self-confidence and can perform better on defense [10]. Previous studies had also shown that steals [41, 42] and blocks [5] were the result of pressure defense and played a crucial role in a team's victory. Moreover, a study regarding female professional basketball players presented different results than male basketball players in that successful 2-point shooting, free throw shooting percentage, and assists were the strongest performance indicators to discriminate starting and bench players in the women's national basketball association league (WNBA) [10].

## Performance indicators of starting and bench players that discriminate between the top and bottom teams

Quantitative analysis of basketball performance through game-related statistics such as shooting percentage, rebounds, and assists has been widely used to measure player performance and analyze game events [28]. In this sense, which game-related statistics can discriminate winning and losing teams had become a hot topic for team performance studies in basketball [43–46]. Basketball games are won or lost by the number of points scored. Shooting is one of the most fundamental basketball skills, and a player's ability to shoot, that is, the percentage of shots taken on the court, is a key indicator of a team's success [47]. This view was consistent with the results of this study. In addition, 2P% of the top four teams was more than 50% in the 2014 Men's Basketball World Cup [47], which was similar to the top four teams in the Tokyo Olympics, illustrating the importance of shooting percentage for successful teams [48]. Free throws are the easiest way to score, and national teams seem to place more emphasis on free throw shooting. Previous studies had shown that the top teams were hitting 73 percent or more of their free throws during the 2019 Men's Basketball World Cup [47], which was consistent with the top four teams at the Tokyo Olympics. However, there were different findings in the study of the Turkish men's basketball league, which concluded that there was no difference in terms of successful 2-point and 3-pointers, and successful free throws between the top and bottom teams [43]. The inconsistency that caused the results of the study may be due to different levels of matches, or different national leagues.

Some limitations should be acknowledged when interpreting the results of this study. First of all, there were other variables that may affect a team's starting lineup during the Olympic basketball tournament, resulting in a starting lineup that may not always be the strongest five players. The most common of these may be affected by injuries to starters and may also be due to a team's particular tactical arrangements. For example, if a team has advanced to the next round in the group stage of the Olympics, it may adjust its starting lineup for the rest of the games at this stage, with the aim of resting its starters as much as possible or avoiding injuries. However, these influences were beyond the control of this study and depended largely on the decisions of the coaches. Another limitation was that this study analyzed data solely from the Tokyo Olympics to determine the attack-defense ability of starting and bench players for each team. However, it could be that some players may not play at their true ability levels for various reasons in this competition. Therefore, the analysis of multiple elite-level basketball competitions can more fully reflect the attack-defense ability of the team's starting and bench players in future studies.

## Conclusion

Based on the results of this study, one of the common characteristics of the top national basketball teams during the Tokyo Olympics men's basketball tournament was the strong attack-defense ability of the starting players, mainly in terms of 3P% and FT%. Therefore, for those national teams at the bottom of the rankings, it was recommended to strengthen 3-point shooting and free-throw shooting practice for starting players. Or select players with stronger 3-point shooting ability and more accurate free-throw shooting into the starting rotation to improve the overall attack-defense strength of the team. Also, the ability to shoot accurate 2-pointers and actively share the ball through good teamwork were powerful tools to improve a team's attack-defense performance. Bench players should pay attention to improving their 3-point shooting technique, enhancing their rebounding awareness, and imposing pressure defense on opponents to increase the chances of steals and blocks so that they may be able to make the team's starting lineup. The results seem to be very valuable for coaches to clarify the

attacking and defensive levels of starting and bench players, and to discover the strengths and weaknesses of different role players in terms of technical indicators, so as to make reasonable player rotation decisions for future games, as well as provide targeted training to improve the overall performance of the team.

## Author Contributions

**Conceptualization:** Wenping Sun, ChenSoon Chee.

**Data curation:** LianYee Kok, FongPeng Lim, Shamsulariffin Samsudin.

**Formal analysis:** ChenSoon Chee.

**Methodology:** FongPeng Lim.

**Supervision:** ChenSoon Chee, LianYee Kok.

**Validation:** Shamsulariffin Samsudin.

**Writing – original draft:** Wenping Sun.

**Writing – review & editing:** LianYee Kok.

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
