## [Decision Letter · Decision Letter 0]

30 Aug 2023

PONE-D-23-16716Differentiating attack-defense performance for starting and bench players during the Tokyo Olympics men's basketball competitionPLOS ONE

Dear Dr. sun,

Thank you for submitting your manuscript to PLOS ONE. After careful consideration, we feel that it has merit but does not fully meet PLOS ONE’s publication criteria as it currently stands. Therefore, we invite you to submit a revised version of the manuscript that addresses the points raised during the review process.

We look forward to receiving your revised manuscript.

Kind regards,

Sebastián Del Rosso, PhD

Guest Editor

PLOS ONE

Journal Requirements:

Reviewers' comments:

Reviewer's Responses to Questions

**Comments to the Author**

1. Is the manuscript technically sound, and do the data support the conclusions?

Reviewer #1: Yes

Reviewer #2: Partly

2. Has the statistical analysis been performed appropriately and rigorously? 

Reviewer #1: Yes

Reviewer #2: No

3. Have the authors made all data underlying the findings in their manuscript fully available?

Reviewer #1: Yes

Reviewer #2: Yes

4. Is the manuscript presented in an intelligible fashion and written in standard English?

Reviewer #1: Yes

Reviewer #2: Yes

5. Review Comments to the Author

Reviewer #1: The authors examined an interesting topic that has a solid practical application value within a sports setting. Please find below my comments and suggestions for improvement.

Line 19 – eliminate (p<0.05) since you already stated the level of statistical significance in the previous sentence.

Line 22 – “points per game” instead of “point per game”. Also, you should just say “… statistically significant and strong positive correlation…” instead of “very significant…”

Line 26-27 – I would suggest authors to rephrase this sentence. Instead of “telling coaches what to do”, you may want to modify this to sounds more like recommendation/suggestion.

Line 31 – “situation on the court” should be substituted with “on-court competitive demands”

Line 32 – “in time” – “in timely manner”

Line 45 – I would suggest authors to include the below listed references here to further support this statement pertaining to game-related statistics.

• Cabarkapa, D., Deane, M. A., Fry, A. C., Jones, G. T., Cabarkapa, D. V., Philipp, N. M., & Yu, D. (2022). Game statistics that discriminate winning and losing at the NBA level of basketball competition. Plos One, 17(8), e0273427.

• Csataljay, G., O’Donoghue, P., Hughes, M., & Dancs, H. (2009). Performance indicators that distinguish winning and losing teams in basketball. International Journal of Performance Analysis in Sport, 9(1), 60-66.

• Cabarkapa, D., Deane, M. A., Cabarkapa, D. V., Jones, G. T., & Fry, A. C. Differences in game-related statistics between winning and losing teams in NCAA Division-II men’s basketball. Journal of Applied Sports Sciences, 2, 3-10.

• Trninić, S., Dizdar, D., & Lukšić, E. (2002). Differences between winning and defeated top quality basketball teams in final tournaments of European club championship. Collegium Antropologicum, 26(2), 521-531.

Line 54 – Please rephrase this sentence. For example, “Currently, there is a lack of scientific literature focused on examining..."

Line 60-61 – You can just say “for starting and bench players, as well as to discover common…”

Line 71 – Spell out USA before you use the abbreviation for the first time.

Line 73 – This “i.e., starting and bench players” should be in parenthesis.

Line 80 – Correct reference format

Line 86 – Did you use any data scraping software to obtain the data or this was done by hand?

Line 113 – List SPSS software manufacturer in parenthesis.

Line 132 – I don’t think that word “obviously” is necessary here. It should be eliminated.

Table 2 – I think that the outline of this table needs to be improved. You need to have a long table that may spread across a full page. The data should not be separated in two different sections.

Table 3 – Same comment as above.

Line 163 – “weak” term should be modified/substituted. Maybe “not successful as other countries…” or similar.

186 – I believe that you did not use PPG, 2P%, etc. abbreviations earlier in the text. Please keep the terminology consistent and define abbreviations before using them.

Table 5 – Include definitions of abbreviations under the table as a note.

Table 7 – Shouldn’t Pearson (p) be Pearson (r)?

Line 238 – I think that the second part of the sentence may be too strong of a statement “…and because of that they won a championship”. In my opinion, this sentence can be modified.

Line 266 – You say “there are a couple of research reports that used game-related…”, but you do not cite them. Please include the appropriate citations.

Line 278 – Jumping ability is one of the required physical performance parameters for successful rebounding in basketball. However, there are other, such as perception, or ability to predict where the ball will bounce. For example, something that Denis Rodman was very good at. Maybe that might be worth mentioning that jumping ability is not the only important thing.

Line 280 – You are discussing the impact of shooting distance here. You might want to include some of the below listed references and further support or briefly explain how an increase in distance can later shooting accuracy.

• Okazaki, V. H., Rodacki, A. L., & Satern, M. N. (2015). A review on the basketball jump shot. Sports biomechanics, 14(2), 190-205.

• Cabarkapa, D., Cabarkapa, D. V., Philipp, N. M., Eserhaut, D. A., Downey, G. G., & Fry, A. C. (2022). Impact of distance and proficiency on shooting kinematics in professional male basketball players. Journal of Functional Morphology and Kinesiology, 7(4), 78.

Line 310 – You had 3P% earlier, not you use “3-point shooting percentage”. Please make sure that the terminology remain identical, including abbreviations, throught the overall manuscript.

Line 315 – Please check grammar. Also, I assume that you refer to coaches here. Please specify.

Reviewer #2: The objective of this study was to assess the differences in attack-defense performance between the top and bottom teams for the starting players, as well as for the bench players, to discover common characteristics of the top basketball teams. Also, to determine the relationship between the attack-defense performance of the starting and bench players and the final competition rankings, as well as with each attack-defense performance indicator. The study presents some interesting findings and an interesting approach to analyzing basketball performance. However, there are several points that need to be addressed properly. For instance, what about the potential influence of confounders, such as the injury rate, the changes in the game plan according to the opponent, or the physical fitness of players? Also, during the tournament, there is a group phase, a cross-group classification, and the playoffs. In this regard, according to the results obtained by a team, there could be several changes in the team’s playing strategy leading to changes in the starting formation. Moreover, given the explanation provided by the authors, I am not sure about the comparison between the top four and the bottom four teams. Considering all this, I’ll suggest trying to consider any potential confounder (if possible) to isolate those variables of interest.

Introduction: The introduction is well-written and leads to the objectives and hypotheses.

Methods: The methods section needs to address more in deep the analysis performed adding clarity to the obtained results. There are paragraphs in the Results section that should be addressed in this section. In addition, please add a statistical analysis subsection.

Results: The results section needs some polish. There are paragraphs where the authors are repeating information provided in the Tables which is considered redundant information. As a suggestion, part or even all data presented in Tables 2, 3, and 5 could be summarized in graphics which would be much easier to interpret. In addition, there are parts of the results section that should be moved into the Methods section, making the results section more readable and direct.

Discussion: Although the discussion is founded on the data and results, it could benefit from a deeper discussion of some topics to reinforce the findings.

SPECIFIC COMMENTS.

L18: The word “showed” appears two times in the same sentence, could you use a synonym?

L22: Although the term “very significant” could be used to indicate very low P-values is a vague term statistically speaking. Preferably, indicate the actual P-values and correlations or alternatively include the range of correlations and P-values.

L49-50: “While the different results were successful…” I suggest rewriting this sentence to differentiate this study from the previous one.

L51: “Integrated” should be “integrating”

L53: Please cite the studies you are referring to.

L57: Maybe a more suitable expression would be “this study designed to analyze and evaluate”

L66: I think it is “was related”, as you are referring to the performance.

L68: It should be “Materials and Methods”

L70: “Game” should be “tournament”. Also, please review the wording for “In order of final ranking.”

L73: What do you mean by “as the sample”

L103: Tables should be cited in order of appearance. Here, Tables 2 and 3 are mentioned before Table 1. Please correct.

L113: Have you tested the assumptions for a T-test (i.e., normality of distributions and homogeneity of variance)?

L119-122: The authors should explain, in detail, how you used the correlation coefficients to determine the contribution of different performance indicators to the attack-defense performance of starting and bench players.

L126-127: The authors need to rephrase this sentence as is hard to read.

Table 3: Please include the legend for the acronyms used (as in Table 2).

L160: I am not sure about starting a sentence with “As we all know”

L160-174: This information should be moved to the Methods section. Moreover, I am not sure about this comparison. As the authors mentioned the bottom four teams are generally teams coming from continents with a low basketball tradition. This is due to the classification system of the Olympics games. Thus, there is a high chance that some countries left out of the Olympic competition have better performance indicators than those in the bottom four of the Olympic tournament which in your study would result in a biased analysis that cannot be extrapolated.

Table 4. I do not think this comparison deserves a table. This result can easily be included in the text.

L193-194: Given your results, it cannot be said that the starting players of the top four teams performed better than the bottom four teams in all the indicators.

L196-197: Again, although this sentence is true in numerical values is not true in statistical terms.

L229-235: Please be careful with the use of statistical language. A very significant is a subjective term and does not represent a statistical meaning.

L238: This is a bold statement. I’ll be extremely cautious with this kind of assumption based on post hoc analysis.

L237-248: This paragraph is simply repeating the data already presented in the results section.

L271-273: This sentence seems somewhat speculative. Could you support it with a citation?

L275: “while this difference should not be apparent among national team players”. Again, do you have any data supporting this assumption?

L282: “by a study” or “in a study of the NCAA”

6. PLOS authors have the option to publish the peer review history of their article (what does this mean?). If published, this will include your full peer review and any attached files.

Reviewer #1: No

Reviewer #2: No

---

## [Author Response · Author response to Decision Letter 0]

15 Oct 2023

Thank you for inviting us to submit a revised draft of our manuscript entitled, " Differentiating attack-defense performance for starting and bench players during the Tokyo Olympics men's basketball competition" to PLOS ONE. We also appreciate the time and effort you and each of the reviewers have dedicated to providing insightful feedback on ways to strengthen our paper. Thus, it is with great pleasure that we resubmit our article for further consideration. We have incorporated changes that reflect the detailed suggestions you have graciously provided. We also hope that our edits and the responses we provide below satisfactorily address all the issues and concerns the reviewers have noted.

To facilitate your review of our revisions, the following is a point-by-point response to the questions and comments delivered in your letter.

Review Comments to the Author

REVIEWER 1 COMMENTS:

Reviewer #1: The authors examined an interesting topic that has a solid practical application value within a sports setting. Please find below my comments and suggestions for improvement.

Response: We greatly appreciate the time and effort you put into our manuscript, as well as your valuable comments.

Line 19 – eliminate (p<0.05) since you already stated the level of statistical significance in the previous sentence. 

Response: Thank you for this suggestion. We have replaced p<0.05 with an actual p-value to indicate the level of statistical difference more clearly in our revised manuscript (see line 19-20).

Line 22 – “points per game” instead of “point per game”. Also, you should just say “… statistically significant and strong positive correlation…” instead of “very significant…”.

Response: Thank you for this suggestion. We have made correction according to your comments in our revised manuscript (see line 22-23).

Line 26-27 – I would suggest authors to rephrase this sentence. Instead of “telling coaches what to do”, you may want to modify this to sounds more like recommendation/suggestion.

Response: Thank you for this suggestion. We have re-written this sentence according to your suggestion in our revised manuscript (see line 27-28).

Line 31 – “situation on the court” should be substituted with “on-court competitive demands”

Response: Thanks for this suggestion. As your suggested that we have corrected accordingly in our revised manuscript (see line 32).

Line 32 – “in time” – “in timely manner”

Response: Thanks for this suggestion. We have corrected accordingly in our revised manuscript (see line 33).

Line 45 – I would suggest authors to include the below listed references here to further support this statement pertaining to game-related statistics.

Response: Thanks for your suggestion. We have cited those references you suggested in the revised manuscript (see line 47), and it is really true that these references can further support that statement pertaining to game-related statistics. We believe this will further strengthen the credibility and scientific rigor of our paper. At the same time, we have added these references in the reference section (see line 394-403).

Line 54 – Please rephrase this sentence. For example, “Currently, there is a lack of scientific literature focused on examining..."

Response: Thanks for this suggestion. We have re-written this sentence in our revised manuscript (see line 56-59).

Line 60-61 – You can just say “for starting and bench players, as well as to discover common…”

Response: Thanks for this suggestion. We have corrected accordingly in our revised manuscript (see line 63-64).

Line 71 – Spell out USA before you use the abbreviation for the first time.

Response: Thanks for this suggestion. Here, we have used the full name instead of USA in our revised manuscript (see line 73).

Line 73 – This “i.e., starting and bench players” should be in parenthesis.

Response: Thanks for this suggestion. We have corrected accordingly in our revised manuscript (see line 75-76).

Line 80 – Correct reference format

Response: Thank you for pointing out this error. We have revised the citation format in our revised manuscript (see line 83).

Line 86 – Did you use any data scraping software to obtain the data or this was done by hand?

Response: Thanks for your question. First, we needed to identify the team's starting and bench players, and then extract specific data (e.g., how many assists) for each player, which was done manually based on the information provided by the FIBA website (i and ii of the data collection process, see line 89-91). Secondly, we utilized Microsoft Excel to go through the calculations to obtain the data needed for this study, such as calculating the mean assists for each player in this tournament, as well as calculating the mean assists for all of the starting players on the team, etc. (iii and iv of the data collection process, see line 91-93). Of course, the data will be double-checked to ensure its accuracy.

Line 113 – List SPSS software manufacturer in parenthesis.

Response: Thanks for this suggestion. We have added the manufacturer of SPSS software in our revised manuscript (see line 124).

Line 132 – I don’t think that word “obviously” is necessary here. It should be eliminated.

Response: Thanks for this suggestion. We have removed this word in our revised manuscript (see line 132).

Table 2 – I think that the outline of this table needs to be improved. You need to have a long table that may spread across a full page. The data should not be separated in two different sections.

Table 3 – Same comment as above.

Response: Thanks for this suggestion and comment. Considering your suggestion, we have been changed the layout of Table 2 and 3 to spread across a full page in our revised manuscript (see line 148 and 157).

Line 163 – “weak” term should be modified/substituted. Maybe “not successful as other countries…” or similar.

Response: Thanks for this suggestion. However, we have deleted the paragraph in line 160 of the original manuscript after our further discussion because the statement in this paragraph seems to be unnecessary for this study.

186 – I believe that you did not use PPG, 2P%, etc. abbreviations earlier in the text. Please keep the terminology consistent and define abbreviations before using them.

Response: Thanks for your comment. We have added the corresponding abbreviation where the full name of the term first appears to keep the terminology consistent (see line 84-87).

Table 5 – Include definitions of abbreviations under the table as a note.

Response: Thanks for this suggestion. We have added notes on abbreviations terms below this table in our revision (see line 183-185).

Table 7 – Shouldn’t Pearson (p) be Pearson (r)?

Response: Thank you for pointing out this error. Yes, Pearson (p) has been corrected to Pearson (r) in our revised manuscript (see Table 6).

Line 238 – I think that the second part of the sentence may be too strong of a statement “…and because of that they won a championship”. In my opinion, this sentence can be modified.

Response: Thanks for this suggestion and comment. We have re-written this sentence in our revised manuscript (see line 236-237).

Line 266 – You say “there are a couple of research reports that used game-related…”, but you do not cite them. Please include the appropriate citations.

Response: Thanks for this comment. We have added the corresponding citations after this sentence in our revised manuscript (see line 264).

Line 278 – Jumping ability is one of the required physical performance parameters for successful rebounding in basketball. However, there are other, such as perception, or ability to predict where the ball will bounce. For example, something that Denis Rodman was very good at. Maybe that might be worth mentioning that jumping ability is not the only important thing.

Response: Thanks for this comment. As you mentioned, a player's ability to jump is only one of the elements for successful rebounding in basketball, and it does get affected by other factors as well. We have reviewed the literature again, and found that anticipating where the ball is going to fall, as you said, was also an important factor, as well as a number of other possible factors, which we have included in the revised manuscript (see line 279-281).

Line 280 – You are discussing the impact of shooting distance here. You might want to include some of the below listed references and further support or briefly explain how an increase in distance can later shooting accuracy.

Response: Thanks for this suggestion and comment. Considering your suggestion, we have carefully reviewed the two references you mentioned and found that they can indeed support this idea discussed in our study, so we have cited the two references in our revised manuscript (see line 286-288). At the same time, we have added these references in the reference section (see line 427-431). 

Line 310 – You had 3P% earlier, not you use “3-point shooting percentage”. Please make sure that the terminology remain identical, including abbreviations, throught the overall manuscript.

Response: Thanks for this suggestion and comment. Here, we have corrected “3-point shooting percentage” and “free throw shooting percentage” to “3P%” and “FT%” respectively in our revision (see line 327). Additionally, we double-checked the terminology one more time to ensure consistency throughout the article.

Line 315 – Please check grammar. Also, I assume that you refer to coaches here. Please specify.

Response: Thanks for this comment. We have re-written this sentence in our revised manuscript (see line 331-333).

Special thanks to you for your good comments and suggestions.

REVIEWER 2 COMMENTS:

Reviewer #2: The objective of this study was to assess the differences in attack-defense performance between the top and bottom teams for the starting players, as well as for the bench players, to discover common characteristics of the top basketball teams. Also, to determine the relationship between the attack-defense performance of the starting and bench players and the final competition rankings, as well as with each attack-defense performance indicator. The study presents some interesting findings and an interesting approach to analyzing basketball performance. However, there are several points that need to be addressed properly. For instance, what about the potential influence of confounders, such as the injury rate, the changes in the game plan according to the opponent, or the physical fitness of players? Also, during the tournament, there is a group phase, a cross-group classification, and the playoffs. In this regard, according to the results obtained by a team, there could be several changes in the team’s playing strategy leading to changes in the starting formation. Moreover, given the explanation provided by the authors, I am not sure about the comparison between the top four and the bottom four teams. Considering all this, I’ll suggest trying to consider any potential confounder (if possible) to isolate those variables of interest.

Response: We greatly appreciate the time and effort you put into our manuscript, as well as your valuable comments.

We agree with what you mentioned above that a team's starting lineup does change due to player injuries, player performance, opponent strength or game strategy, so it is not necessarily fixed. Therefore, a team's starting lineup in some matches may not be the strongest five players. This situation may be more common in national basketball leagues because the time span of basketball leagues is generally longer, like the NBA league where the regular season lasts for six months and each team has to play more than 80 games, and such a long period of time and high-intensity matches may be more likely to lead to player fatigue and thus cause injuries. However, a tournament system like the Olympics lasts for a shorter period of time and a team may play up to 6 games in that period. As a result, the chances of players getting injured are likely to be much less than in the league, and the starting lineups are relatively more stable. Nonetheless, we found that some teams did adjust their starting lineups by some of the confounding factors above in this study, but these are currently out of our control and depend largely on the coaches' decisions. 

Therefore, we think this may be one of the limitations for this study, and we have added a paragraph to state this limitation in our revised manuscript (see line 312-322). Also, we look forward to conducting further research in this area.

Introduction: The introduction is well-written and leads to the objectives and hypotheses.

Response: Thanks for this comment.

 Methods: The methods section needs to address more in deep the analysis performed adding clarity to the obtained results. There are paragraphs in the Results section that should be addressed in this section. In addition, please add a statistical analysis subsection.

Response: Thanks for this comment and suggestion. We have made many corrections to Methods section according to your suggestion in our revised manuscript. As you suggested that we have added a subsection on data analysis and focused on explaining the data analysis session in more detail. In addition, some of the content in the results section has been moved to the methods section. Please refer to the detailed responses to specific comments below for specific changes.

Results: The results section needs some polish. There are paragraphs where the authors are repeating information provided in the Tables which is considered redundant information. As a suggestion, part or even all data presented in Tables 2, 3, and 5 could be summarized in graphics which would be much easier to interpret. In addition, there are parts of the results section that should be moved into the Methods section, making the results section more readable and direct.

Response: Thanks for this comment and suggestion. We have made many corrections to the Results section according to your suggestion in our revised manuscript. We have removed some redundant information and adjusted the layout of Tables 2 and 3 to spread across a full page to make them easier to interpret and understand. In addition, as you suggested that some of the content in the results section has been moved to the Methods section. Please refer to the detailed responses to specific comments below for specific changes.

Discussion: Although the discussion is founded on the data and results, it could benefit from a deeper discussion of some topics to reinforce the findings.

Response: Thanks for this comment. Yes, as you say, a more in-depth discussion can further strengthen the findings. Therefore, more references related to this study were reviewed and compared with the results of this study to find out their similarities and differences for more in-depth discussion and analysis. Please refer to the discussion section in our revised manuscript for specific changes.

SPECIFIC COMMENTS：

L18: The word “showed” appears two times in the same sentence, could you use a synonym?

Response: Thanks for this suggestion. Yes, we have replaced "showed" in our revised manuscript (see line 18).

L22: Although the term “very significant” could be used to indicate very low P-values is a vague term statistically speaking. Preferably, indicate the actual P-values and correlations or alternatively include the range of correlations and P-values.

Response: Thanks for this suggestion. As you suggested that we have included the range of correlations and the actual P-values in our revised manuscript (see line 19-23).

L49-50: “While the different results were successful…” I suggest rewriting this sentence to differentiate this study from the previous one.

Response: Thanks for this suggestion. We have re-written this sentence according to your suggestion in our revised manuscript (see line 51-53).

L51: “Integrated” should be “integrating”

Response: Thanks for this suggestion. We have corrected accordingly in our revised manuscript (see line 53).

L53: Please cite the studies you are referring to.

Response: Thanks for this suggestion. We have added the corresponding citations after this sentence in our revised manuscript (see line 56).

L57: Maybe a more suitable expression would be “this study designed to analyze and evaluate”

Response: Thanks for this suggestion. We have corrected accordingly in our revised manuscript (see line 60).

L66: I think it is “was related”, as you are referring to the performance.

Response: Thanks for this comment. Yes, we have corrected accordingly in our revised manuscript (see line 69).

L68: It should be “Materials and Methods”

Response: Thanks for this suggestion. We have changed “Method” to “Materials and Methods” in our revision (see line 71).

L70: “Game” should be “tournament”. Also, please review the wording for “In order of final ranking.”

Response: Thanks for this suggestion. We have corrected accordingly and re-written the first two sentences in our revision (see line 73-74).

L73: What do you mean by “as the sample”

Response: Thanks for this question. This sentence has been rewritten in the revised version so that it can be understood more clearly (see line 75-76).

L103: Tables should be cited in order of appearance. Here, Tables 2 and 3 are mentioned before Table 1. Please correct.

Response: Thanks for this suggestion. It is really true as you suggested that Tables 2 and 3 are mentioned before Table 1. We have deleted "(refer to R values in Tables 2–3)" in the revised version (line 99). This is because R is explained below (see line 99-100) and in more detail on lines 102-104. With these changes, the problem you mentioned about the tables not being cited in order has been solved.

L113: Have you tested the assumptions for a T-test (i.e., normality of distributions and homogeneity of variance)?

Response: Thanks for this question. Yes, we have tested that all data used for independent sample T-test were normally distributed. Also, we have stated this in the revised manuscript (see line 127-128).

L119-122: The authors should explain, in detail, how you used the correlation coefficients to determine the contribution of different performance indicators to the attack-defense performance of starting and bench players.

Response: Thanks for this comment. A detailed explanation that how to use Pearson Correlation was presented in the results section (see line 213-217). Also, we have re-written that sentence you mentioned in the revised manuscript (see line 132-134) so that it can be better understood. 

L126-127: The authors need to rephrase this sentence as is hard to read.

Response: Thanks for this suggestion. We have re-written this sentence in our revised manuscript (see line 138-139) so that it can be easy to understand. 

Table 3: Please include the legend for the acronyms used (as in Table 2).

Response: Thanks for this suggestion. As you suggested that we have added notes on abbreviated terms below Tables 3, 4 and 6 in the revised version.

L160: I am not sure about starting a sentence with “As we all know”

Response: Thanks for this comment. However, we have deleted the paragraph in line 160 of the original manuscript after our further discussion because the statement in this paragraph seems to be unnecessary for this study.

L160-174: This information should be moved to the Methods section. Moreover, I am not sure about this comparison. As the authors mentioned the bottom four teams are generally teams coming from continents with a low basketball tradition. This is due to the classification system of the Olympics games. Thus, there is a high chance that some countries left out of the Olympic competition have better performance indicators than those in the bottom four of the Olympic tournament which in your study would result in a biased analysis that cannot be extrapolated.

Response: Thanks for this suggestion and comment. Considering your suggestion above, we have made the following corresponding changes after our serious discussions.

We have deleted the paragraph in line 160 of the original manuscript, as the statement in this paragraph seems to be unnecessary after our discussion. In addition, we have moved some content of the second paragraph in line 167 to the data analysis section and explained it in further detail in the revised version (see line 117-123).

Furthermore, we agree with you that due to the selection system of the Olympic Games (each continent has a limited number of places in the Olympics), some countries that are stronger than the bottom four teams will not be able to participate in the Olympics. However, the teams that can finish in the top four in the Olympics should be at the top level of the world. Therefore, we have chosen to compare the top teams with the bottom teams in our study, hoping to find out the gap between them in terms of technical indicators, as well as some common characteristics of the top teams, with a view to helping to improve the strength of the bottom teams.

Of course, the top teams in each Olympic Games or Basketball World Cup may be different, or some teams may not play to their true strength in one tournament for some reasons. With in this mind, we intend to analyze multiple international high-level competitions in future study to make the results more comprehensive and scientific. 

Table 4. I do not think this comparison deserves a table. This result can easily be included in the text.

Response: Thanks for this suggestion and comment. Considering your suggestion above, we have removed Table 4 and the result in Table 4 was included in the text in the revised version (see line 171-173). 

L193-194: Given your results, it cannot be said that the starting players of the top four teams performed better than the bottom four teams in all the indicators.

Response: Thanks for this comment. Yes, as mentioned in our study, the starting players of the bottom four teams outperformed the top four teams in terms of steals and fouls, but no significant difference (see line 191-192). Even as the top teams in the world, it is normal that they may have shortcomings in some specific attacking and defensive performance indicators. Therefore, the stating players of the top four teams may need to improve in steals and fouls in the future.

L196-197: Again, although this sentence is true in numerical values is not true in statistical terms.

Response: Thanks for this comment. We have re-written this sentence in our revised manuscript (see line 191-192).

L229-235: Please be careful with the use of statistical language. A very significant is a subjective term and does not represent a statistical meaning.

Response: Thanks for this suggestion and comment. We have re-written this paragraph using standardized statistical language in our revised manuscript (see line 225-233).

L238: This is a bold statement. I’ll be extremely cautious with this kind of assumption based on post hoc analysis.

Response: Thanks for this suggestion and comment. As you suggested that we also realized that this sentence may be too strong of a statement. Therefore, we have modified this sentence in the revised version (see line 236-237).

L237-248: This paragraph is simply repeating the data already presented in the results section.

Response: Thanks for this comment. However, we believe that this paragraph is not just a repetition of the results section, but a summary and generalization of the findings, in order to further develop the detailed discussion section.

L271-273: This sentence seems somewhat speculative. Could you support it with a citation?

L275: “while this difference should not be apparent among national team players”. Again, do you have any data supporting this assumption?

Response: Thanks for these comments. Since the two comments above are similar questions, we'll respond to them together here. In fact, these statements are an inferential analysis based on the results of previous studies and are not an unfounded speculation. However, this assumption needs to be confirmed in future studies. We have re-written these statements in a revised version to make the study more rigorous (see line 268-277).

L282: “by a study” or “in a study of the NCAA”

Response: Thanks for this suggestion. We have corrected accordingly in our revised manuscript (see line 284)

Special thanks to you for your good comments and suggestions.

We tried our best to improve the manuscript and made some changes in the manuscript. These changes will not influence the content and framework of the paper. And here we did not list the changes but marked in red in revised manuscript. 

We appreciate for Editors/Reviewers’ warm work earnestly, and hope that the correction will meet with approval.

Once again, thank you for giving us the opportunity to strengthen our manuscript with your valuable comments and queries.

---

## [Decision Letter · Decision Letter 1]

14 Nov 2023

PONE-D-23-16716R1Differentiating attack-defense performance for starting and bench players during the Tokyo Olympics men's basketball competitionPLOS ONE

Dear Dr. sun,

Thank you for submitting your manuscript to PLOS ONE. After careful consideration, we feel that it has merit but does not fully meet PLOS ONE’s publication criteria as it currently stands. Therefore, we invite you to submit a revised version of the manuscript that addresses the points raised during the review process.

We look forward to receiving your revised manuscript.

Kind regards,

Sebastián Del Rosso, PhD

Guest Editor

PLOS ONE

Journal Requirements:

Reviewers' comments:

Reviewer's Responses to Questions

**Comments to the Author**

1. If the authors have adequately addressed your comments raised in a previous round of review and you feel that this manuscript is now acceptable for publication, you may indicate that here to bypass the “Comments to the Author” section, enter your conflict of interest statement in the “Confidential to Editor” section, and submit your "Accept" recommendation.

Reviewer #1: All comments have been addressed

Reviewer #2: (No Response)

2. Is the manuscript technically sound, and do the data support the conclusions?

Reviewer #1: Yes

Reviewer #2: Yes

3. Has the statistical analysis been performed appropriately and rigorously? 

Reviewer #1: Yes

Reviewer #2: Yes

4. Have the authors made all data underlying the findings in their manuscript fully available?

Reviewer #1: Yes

Reviewer #2: Yes

5. Is the manuscript presented in an intelligible fashion and written in standard English?

Reviewer #1: Yes

Reviewer #2: Yes

6. Review Comments to the Author

Reviewer #1: (No Response)

Reviewer #2: I thank the authors for their efforts to respond to each of my comments, you have done a very good job. I have just some minor comments for the last version of the manuscript.

There are some minor English language issues in the manuscript. Please revise it.

In the statistical analysis section. Please state the analysis for testing the normality of the variable’s distribution. Also, you need to acknowledge which test you have used to analyze the homogeneity of variance for the independent t-tests. Finally, unless otherwise, you only need to mention the alpha level once (there is no need to mention the significance level for each test).

L132-134: This paragraph is a bit odd. Could you rephrase it?

In the results section, you describe again the comparisons and correlations procedures (e.g., L168, L176, and L198). In the results section, the reader is expecting the results of your analysis and not the description of it. Thus, I would suggest describing comprehensively your statistical analysis and comparisons in the proper section to avoid repeating information.

7. PLOS authors have the option to publish the peer review history of their article (what does this mean?). If published, this will include your full peer review and any attached files.

Reviewer #1: No

Reviewer #2: No

---

## [Author Response · Author response to Decision Letter 1]

1 Dec 2023

Thank you for inviting us to resubmit a revised draft of our manuscript entitled, "Differentiating attack-defense performance for starting and bench players during the Tokyo Olympics men's basketball competition" to PLOS ONE. We also appreciate the time and effort you and each of the reviewers have dedicated to providing insightful feedback on ways to strengthen our paper. We have incorporated changes that reflect the detailed suggestions you have graciously provided. We also hope that our edits and the responses we provide below satisfactorily address all the issues and concerns the reviewers have noted.

To facilitate your review of our revisions, the following is a point-by-point response to the questions and comments delivered in your letter.

Review Comments to the Author

Reviewer #2: I thank the authors for their efforts to respond to each of my comments, you have done a very good job. I have just some minor comments for the last version of the manuscript.

Response: We greatly appreciate the time and effort you put into our manuscript, as well as your valuable comments.

There are some minor English language issues in the manuscript. Please revise it.

Response: Thank you for this suggestion. Some English language issues have been corrected in our revised manuscript (see line 32-33,41,57,60,64,69,89,101-104,113,123,147,174,201,213,246,262,316,326, and 334).

In the statistical analysis section. Please state the analysis for testing the normality of the variable’s distribution. Also, you need to acknowledge which test you have used to analyze the homogeneity of variance for the independent t-tests. 

Response: Thank you for this suggestion. We used Shapiro-Wilk test to test the normality distribution of the dependent variables in this study. Also, Levene’s test was used to test the homogeneity of variance. We have included these in the revised manuscript (see line 127-129). 

Finally, unless otherwise, you only need to mention the alpha level once (there is no need to mention the significance level for each test).

Response: Thank you for this suggestion. Based on your suggestion, we have corrected the relevant paragraph in the revised manuscript (see line 129-130), and addressed the issue you mentioned above.

L132-134: This paragraph is a bit odd. Could you rephrase it?

Response: Thank you for this suggestion. We have re-written this paragraph in our revised manuscript (see line 135-137).

In the results section, you describe again the comparisons and correlations procedures (e.g., L168, L176, and L198). In the results section, the reader is expecting the results of your analysis and not the description of it. Thus, I would suggest describing comprehensively your statistical analysis and comparisons in the proper section to avoid repeating information.

Response: Thank you for this suggestion. It is really true as you mentioned that there is some repeating information in the results section. We have deleted the contents of lines 168, 176, 198 and 213, and also corrected some of the expressions in the revised manuscript (see line 172-173,187,200-201,213-214). Additionally, considering your suggestion, we have detailed the statistical analysis of this study in Data analysis section (see line 124-137), and give the reader a clearer understanding of how each statistical method was applied in this study.

Special thanks to you for your good comments and suggestions.

We appreciate for Editors/Reviewers’ warm work earnestly, and hope that this correction will meet with approval.

Once again, thank you for giving us the opportunity to strengthen our manuscript with your valuable comments and queries.

---

## [Editor Report · Decision Letter 2]

5 Dec 2023

Differentiating attack-defense performance for starting and bench players during the Tokyo Olympics men's basketball competition

PONE-D-23-16716R2

Dear Dr. Sun,

We’re pleased to inform you that your manuscript has been judged scientifically suitable for publication and will be formally accepted for publication once it meets all outstanding technical requirements.

Kind regards,

Sebastián Del Rosso, PhD

Guest Editor

PLOS ONE

---

## [Editor Report · Acceptance letter]

8 Dec 2023

PONE-D-23-16716R2 

 Differentiating attack-defense performance for starting and bench players during the Tokyo Olympics men's basketball competition 

Dear Dr. Sun:

I'm pleased to inform you that your manuscript has been deemed suitable for publication in PLOS ONE. Congratulations! Your manuscript is now with our production department. 

Kind regards, 

on behalf of

Dr. Sebastián Del Rosso 

Guest Editor

PLOS ONE